# Handheld dynamometry: Validity and reliability of measuring hip joint rate of torque development and peak torque

Katherine McNabb[1]*, María B. Sánchez[1], James Selfe[1], Neil D. Reeves[2,3], Michael Callaghan[1,4,5]

1 Department of Health Professions, Faculty of Health and Education, Manchester Metropolitan University, Manchester, United Kingdom, 2 Department of Life Sciences, Faculty of Science and Engineering, Manchester Metropolitan University, Manchester, United Kingdom, 3 Lancaster Medical School, Faculty of Health and Medicine, Lancaster University, Lancaster, United Kingdom, 4 Centre for Musculoskeletal Research, Manchester Academic Health Sciences Centre, University of Manchester, Manchester, United Kingdom, 5 Department of Physiotherapy, Manchester University Hospitals NHS Foundation Trust, Manchester, United Kingdom

☯ These authors contributed equally to this work.
* katherine.mcnabb@stu.mmu.ac.uk

**Data Availability Statement:** All primary data files are available from the https://e-space.mmu.ac.uk/ database (https://doi.org/10.23634/MMU.00634821).

## Abstract

### Introduction

Measuring rate of torque development (RTD) and peak torque (PT) for hip muscle performance presents challenges in clinical practice. This study investigated the construct validity of a handheld dynamometer (HHD) versus an isokinetic dynamometer (IKD), and intra-rater repeated reliability for RTD and PT and their relationship in hip joint movements.

### Methods

Thirty healthy individuals (mean age = 30 ± 8 years, 13 males) underwent two test sessions in a single day. RTD (0–50, 0–100, 0–150, 0-200ms) and PT normalised to body mass in maximal voluntary isometric contractions were measured using a HHD and an IKD in hip flexion, extension, abduction, adduction, internal and external rotation.

### Results

For validity between the devices, $RTD_{0-50}$ exhibited the largest significant systematic bias in all hip movements (3.41–11.99 $Nm \cdot s^{-1} \, kg^{-1}$) and widest limits-of-agreement, while $RTD_{0-200}$ had the lowest bias (-1.33–3.99 $Nm \cdot s^{-1} \, kg^{-1}$) and narrowest limits-of-agreement. For PT, agreement between dynamometers was observed for hip flexion (0.08 $Nm \cdot kg^{-1}$), abduction (-0.09 $Nm \cdot kg^{-1}$), internal (-0.01 $Nm \cdot kg^{-1}$), and external rotation (0.05 $Nm \cdot kg^{-1}$). For reliability, intra-rater intraclass correlation coefficient ($ICC_{2,1}$) ranged from moderate to good in $RTD_{0-50}$ and $RTD_{0-100}$ (0.5–0.88), and good to excellent in $RTD_{0-150}$ and $RTD_{0-200}$ (0.87–0.95) in all movements. The HHD displayed excellent intra-rater, relative reliability values ($ICC_{2,1}$) in all movements (0.85–0.95). Pearson's correlation revealed good linear correlation between PT and $RTD_{0-150}$ and $RTD_{0-200}$ in all movements ($r = .7$ to .87, $p = < .001$).

**Funding:** The author(s) received no specific funding for this work.

**Competing interests:** The authors have declared that no competing interests exist.

## Conclusion

Validity analysis demonstrated significant systematic bias and lack of agreement in RTD measures between the HHD and IKD. However, the HHD displays excellent to moderate intra-rater, relative reliability for RTD and PT measures in hip movements. Clinicians may use the HHD for hip muscle PT assessment but note, late phase RTD measures are more reliable, valid, and relate to PT than early phase RTD. Additionally, the correlation between RTD and PT at various time epochs was examined to better understand the relationship between these measures.

## Introduction

In individuals with hip-related pain, clinicians commonly use peak torque (PT) parameters to assess the maximum force production of a group of muscles, as previous research has observed a reduction in PT measures in several hip muscles during a maximal voluntary isometric contraction (MVIC) in these individuals [1, 2]. Consequently, PT serves as a widely utilized strength metric for clinicians seeking to quantitatively define function and performance, and the effects rehabilitation has in individuals with hip-related pain [3]. However, given that many activities related to sport and daily living do not always demand maximum torque, other research suggests that rate of torque development (RTD) could offer insights into muscle performance prior to reaching PT [4].

RTD is a strength parameter of interest to clinicians and researchers as it provides valuable insights into the neuromuscular characteristics and functional capabilities of individuals. RTD measures how quickly an individual or group of muscles can generate torque during the early phase of contraction [5, 6]. It reflects the performance of fast movements in sporting tasks such as sprinting, jumping, or kicking, as well as in daily functional movements such as correcting balance and changing direction [7].

While these two muscle strength parameters, RTD and PT, offer assessments of distinct aspects of muscle function, PT is known to have a role in determining RTD [6] and therefore establishing their correlation in hip muscle performance is crucial for understanding the interplay between maximum strength and speed of torque generation. Using incremental time epochs to measure RTD allows analysis of various stages in the contraction which are considered to have differing factors influencing the RTD and therefore provide information that could help design targeted rehabilitation to address individual impairments [8, 9].

A laboratory-based isokinetic dynamometer (IKD) is commonly used to measure RTD, but a handheld dynamometer (HHD) can be employed more conveniently in a clinical setting using an isometric contraction. Establishing concurrent validity between the two devices for RTD measurements in hip movements is necessary to give confidence in knowing how accurately the HHD measures the same construct as the IKD. Mentiplay et al. [10] revealed strong concurrent validity for hip flexion, extension, abduction, and adduction with the IKD (RTD ICC ≥0.75) however further research is needed to identify potential systematic bias between these two instruments when measuring RTD in hip movements.

Furthermore, previous research has demonstrated the HHD exhibits strong intra-rater repeated reliability for assessing RTD during hip movements flexion, extension, abduction, and adduction [10–12]. Further investigation is still needed to determine its accuracy in capturing RTD across different time epochs, as reliability varies across epochs ranging from 0-50ms to 0-300ms in these movements.

The reliability and validity of using the HHD to measure peak force in some hip movements has been confirmed in other studies [10, 13–15]. However, as far as the authors are aware, no study has yet investigated the agreement between the HHD with the IKD, and the intra-rater repeated reliability when measuring RTD across six hip movements in three cardinal planes.

The primary objective of this study was to establish the concurrent validity of the HHD against the IKD, and the intra-rater repeated reliability of the HHD when measuring RTD in the time epochs 0–50, 0–100, 0–150 and 0-200ms, and PT, during hip flexion, extension, abduction, adduction, internal and external rotation movements in healthy adults.

## Materials and methods

### Study design

This was an observational study using a test-retest design with two identical test sessions. It was conducted in a movement laboratory at Manchester Metropolitan University, England. All tests were conducted by a single physiotherapist with 25 years clinical experience (first author).

Recommendations from the Guidelines for Reporting Reliability and Agreement Studies [16] were used, and outcome measures recorded were $RTD_{50}$, $RTD_{100}$, $RTD_{150}$, $RTD_{200}$ ($Nm \cdot s^{-1}$) and PT (Nm) for hip flexion, extension, abduction, adduction, internal and external rotation, using an IKD and a HHD.

### Participants

A sample of 30 healthy adults were recruited from university staff, students, friends, and family between 25/9/2020 and the 18/12/2020. Individuals between 18–50 years were eligible for the study with participants comparable in age with individuals experiencing hip-related pain [17]. Exclusion criteria included any self-reported history of hip, groin, lumbar, or lower limb pain or injury that had interfered with function, walking or caused the individual to seek treatment in the preceding 12 months; previous or current significant hip pathology; systemic disease affecting the muscular or nervous systems.

All procedures were approved by the Faculty of Health and Education, ethics committee Manchester Metropolitan University (EthOS ID 11792) and conducted in accordance with the 2013 Helsinki declaration. Informed written consent was obtained from all participants prior to entering the study.

### Instrumentation

A fixed laboratory based IKD (Humac Norm (Cybex), Computer Sports Medicine Inc., Stoughton, MA, USA) and a HHD (Model 01165, Lafayette Instrumentation) were used to record measurements in each test session (Fig 1).

Isometric torque data from the IKD, sampled at 100Hz, was exported to a data acquisition system (Labchart 8, AD Instruments), where the sampling rate was maintained at 100Hz. Additionally, a zero-phase shift low-pass filter with a cut-off frequency of 50 Hz was applied [18]. The HHD data, sampled at 40Hz, was exported to an Excel spreadsheet (Microsoft Excel, USA). Subsequently, numerical data from both devices was manually extracted for analysis. Prior to testing each participant, both dynamometers were calibrated as per manufacturers' instructions.

### Experimental procedure and design

All participants underwent two test sessions, separated by a minimum of 30-minutes. During the first session they were tested first using the IKD, and second with the HHD, to measure

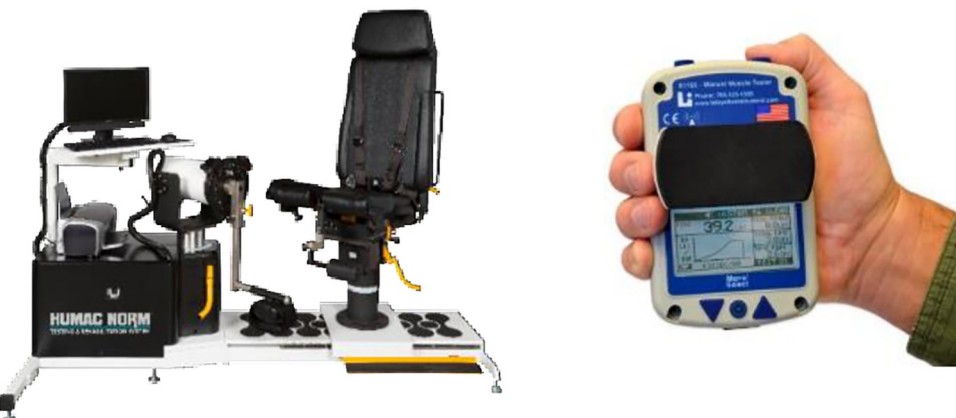

**Fig 1.** Isokinetic dynamometer (left) and Lafayette handheld dynamometer (right). Photos used with permission from copyright owner.

both RTD and PT in the six hip movements. In the second session the measurements were repeated using only the HHD. The order of test positions (Fig 2) was consistent and arranged to prevent individual muscles being tested succinctly with differing hip movements, minimise position changes, and reduce time burden to the participant. Immediately prior to the first test

| | Hip movement position for isokinetic dynamometer and handheld dynamometer testing. | | | | | |
|---|---|---|---|---|---|---|
| | Flexion | Extension | Abduction | Adduction | Internal Rotation | External Rotation |
| Position using IKD | | | | | | |
| Position using HHD | | | | | | |
| Point of HHD application. (Measured to the inferior edge of transducer pad.) | Anterior thigh, 5cm above the superior pole of the patella. | Posterior tibia, in line with 5cm above superior tip of lateral malleolus. | Lateral lower leg, in line with 5cm above superior tip of lateral malleolus. | Medial tibia, in line with 5cm above superior tip of lateral malleolus. | Lateral lower leg, in line with 5cm above superior tip of lateral malleolus. | Medial tibia, in line with 5cm above superior tip of lateral malleolus. |
| Moment arm for HHD | Superior tip of the greater trochanter to 5cm above the superior pole of the patella. | Superior tip of the greater trochanter to 5cm above superior tip of lateral malleolus. | Superior tip of the greater trochanter to 5cm above superior tip of lateral malleolus. | Superior tip of the greater trochanter to 5cm above superior tip of lateral malleolus. | Superior tip of the greater trochanter to lateral knee joint line. | Superior tip of the greater trochanter to lateral knee joint line. |

**Fig 2. Participant's positions used for testing for the isokinetic dynamometer (IKD) and the handheld dynamometer (HHD); point of application and moment arm definition for the HHD.**

session anthropometric measurements were taken and the International Physical Activity Questionnaire (IPAQ), short form was completed to establish participants' physical activity levels. To enable comparison of torque measurements between the IKD and the HHD, the external moment arms around the hip joint were measured in meters (m), using a tape measure (Fig 2) at the start of test session 1. Participants undertook a 5-minute submaximal warm-up on a static exercise bike. Prior to executing the MVIC for testing, participants performed familiarization trials consisting of two submaximal isometric contractions for each hip movement, on both devices [19].

Clinical feasibility of the HHD was a priority and therefore HHD testing positions were selected first and determined based on good reliability results from previous studies (ICC 0.7–0.98) [11, 15, 20, 21]. We aimed to replicate the IKD positions with those of the HHD, but the pre-programmed IKD configuration limited exact replication.

With both dynamometers, four trials of MVIC were performed, with 30 second rest between each trial, and instruction to push as 'hard' as possible for three seconds to record PT. This was followed by a further four trials of MVIC, to measure RTD, with 30 seconds rest between each trial, but with the alternative instruction to push as 'fast and hard' as possible for ~ 1 second to encourage rapid muscle contraction [19, 22]. Countermovement, seen as a movement of the limb in the opposite direction prior to rapid contraction, should be avoided [19] and trials were repeated if this was observed. Participants rested whilst set up was adjusted for the next hip movement. Uniform instruction and standardised verbal encouragement were given throughout [23]. The right limb was tested for all participants.

Participants' measurements with the IKD adhered to the manufacturer's instructions for each hip movement. The IKD rotational axis was visually aligned with the greater trochanter for hip flexion, extension, abduction, and adduction, while alignment for internal and external rotation followed the line of the femur through the anterior knee joint. The transducer pad was securely strapped to the limb.

Following this, participants transitioned to a physiotherapy plinth, where measurements were taken using the HHD using the same contraction protocol as the IKD. Participants optimized stabilization by holding the plinth where required.

### Data extraction

The numerical raw torque data (in Nm) for the IKD was extracted from the Labchart trace. For the HHD, numerical raw force data (in N) was extracted and converted to torque (Nm) with multiplication by the external moment arm length in meters (m).

RTD was calculated as the slope of the torque-time curve (Δ torque / Δ time) [19] and measured as Newton-meters per second (Nm·s$^{-1}$). Each MVIC trial generated four distinct time epochs each starting from the onset of torque production: 0–50, 0–100, 0–150, and 0-200ms [8].

The onset of torque production was determined manually by adding two Newton-meters to the baseline, which was calculated as the mean of three values from the visual flat line over a 500ms period before the observed change in data values [4]. The consistent method of calculation for the onset of torque production was applied for both the HHD and IKD. Numerical torque values were then extracted from the torque trace for the IKD and the Excel spreadsheet for the HHD at 50, 100, 150 and 200ms after torque onset. RTD was calculated for each time epoch. The highest RTD value among the four trials, for each time epoch, was retained for statistical analysis.

For the IKD, PT for each MVIC trial was identified as the highest value observed on visual inspection of the torque trace. For the HHD, PT for each MVIC trial was automatically

generated and displayed as a numerical value in the Excel spreadsheet. The highest PT value among the four MVIC trials was retained for statistical analysis [5]. Results were then normalised to body mass. The outcome variables were, normalised RTD 0–50, 0–100, 0–150, 0-200ms ($Nm \cdot s^{-1} kg^{-1}$) and normalised PT ($Nm \cdot kg^{-1}$).

## Statistical analysis

All statistical analyses were performed in IBM SPSS version 27.0 (IBM Corp. in Armonk, NY). Normality of the data was determined using the Shapiro-Wilk test with the data conforming to normal distribution. Descriptive statistics (means ± standard deviations (SD)) were calculated for all RTD and PT measures for each test session. The mean difference between test sessions 1 and 2 are presented for RTD and PT measurements to allow assessment of systematic bias between sessions.

Bland-Altman analyses with 95% limits of agreement (LOA) were calculated for all measures to examine the systematic difference between the HHD and the IKD. These analyses provide a visualisation of the bias between the measurement methods for each hip movement [24] and determine the agreement between the HHD and IKD systems.

The mean difference (bias) was analysed using a one sample t-test and considered significant if the 95% confidence intervals (95%CI) of the mean difference did not include the line of equality (bias) = 0 [25]. The LOA were calculated using the equation: MD±1.96*SD, where MD is the mean difference between measurement methods and SD the corresponding standard deviation [26].

Intra-rater, test-retest reliability of using the HHD to measure RTD and PT were assessed using the two-way random effects intraclass correlation coefficient ($ICC_{2,1}$), with 95% CI [27]. ICC values were considered excellent ($\geq 0.90$), good (0.89–0.75), moderate (0.74–0.5), or poor ($<0.05$) [28].

The absolute reliability of the HHD RTD and PT was assessed by calculating the standard error of measurement (SEM) using the square root of the mean squared error obtained from a two-way ANOVA. The minimal detectable change (MDC) was calculated using the formula MDC = SEM x 1.96 x $\sqrt{2}$ [27]. Both SEM and MDC were converted to percentages of the mean results.

Pearson correlation was used to assess the correlation between $RTD_{0-50}$, $RTD_{0-100}$, $RTD_{0-150}$, $RTD_{0-200}$ and PT when measured with the IKD.

## Results

All 30 participants data were used in the statistical analysis (mean age = 30 ± 8 years; height = 1.70 ± 0.08m; mass = 70.55 ± 11.76kg), with 13 males included in the study. IPAQ results classified individuals as participating in low [6], moderate [11] and high [13] activity levels.

### Concurrent validity of HHD versus IKD with RTD measures

Bland-Altman analysis showed a significant systematic bias between the HHD and the IKD, when measuring RTD in all six movements, and in all time epochs ($RTD_{0-50}$, $RTD_{0-100}$, $RTD_{0-150}$, $RTD_{0-200}$). The bias and the LOA for RTD are presented in numerical style in Tables 1–4. The mean difference was biased positively indicating measurements from the HHD yielded lower measures than the IKD in all time epochs, and in all hip movements, with one exception of hip extension $RTD_{0-150}$ (-1.33 $Nm \cdot s^{-1} kg^{-1}$).

A noticeable trend was evident in all hip movements, where the late phase time epochs ($RTD_{0-150}$, $RTD_{0-200}$) exhibit better agreement with lower bias and narrower LOA compared to the early phase time epochs ($RTD_{0-50}$, $RTD_{0-100}$).

**Table 1. Intersession reliability of RTD 0-50ms measurements in hip movements using a handheld dynamometer, and agreement analysis with an isokinetic dynamometer.**

| | Intersession reliability | | | | | | | | Agreement analysis | | |
|---|---|---|---|---|---|---|---|---|---|---|---|
| | Test mean(SD) session 1 (Nm·s⁻¹kg⁻¹) | Test mean(SD) session 2 (Nm·s⁻¹kg⁻¹) | Difference(SD) session 1–2 (Nm·s⁻¹kg⁻¹) | $ICC_{2,1}$ | SEM (Nm·s⁻¹kg⁻¹) | SEM % | MDC (Nm·s⁻¹kg⁻¹) | MDC % | Bias (Nm·s⁻¹kg⁻¹) | 95% LOA (Nm·s⁻¹kg⁻¹) | |
| | | | | | | | | | | Lower | Upper |
| **Hip Flexion** | 2.64 (1.21) | 2.48 (1.89) | 0.16 (1.23) | 0.72 | 0.80 | 31 | 1.78 | 70 | 11.99 | -10.0 | 35.0 |
| **Hip Extension** | 3.54 (1.21) | 3.09 (0.95) | 0.45 (1.25) | 0.50 | 0.88 | 27 | 1.97 | 60 | 4.66 | -5.9 | 15.2 |
| **Hip Abduction** | 3.30 (2.06) | 3.00 (1.19) | 0.34 (1.57) | 0.72 | 1.11 | 36 | 2.49 | 80 | 4.67 | -4.9 | 14.2 |
| **Hip Adduction** | 2.89 (1.10) | 3.03 (1.20) | -0.13 (0.96) | 0.79 | 0.68 | 23 | 1.52 | 51 | 3.41 | -2.9 | 9.60 |
| **Hip Internal rotation** | 1.48 (0.76) | 1.42 (0.63) | 0.06 (0.60) | 0.79 | 0.42 | 29 | 0.93 | 64 | 3.67 | -0.9 | 8.27 |
| **Hip External rotation** | 1.40 (0.62) | 1.51 (0.60) | -0.11 (0.47) | 0.82 | 0.33 | 23 | 0.74 | 51 | 6.60 | 0.14 | 13.0 |

SD: Standard deviation, ICC: Intraclass correlation coefficient $ICC_{2,1}$ SEM: Standard error of measurement, MDC: Minimal detectable change, Bias: mean of differences between measurement methods, 95% LOA: 95% Limits of agreement

Bland-Altman plots examining agreement between the HHD and the IKD when measuring hip flexion RTD for each time epoch are shown in Fig 3. As an example of interpretation, the LOA between the HHD and the IKD for measuring RTD 0-50ms in hip flexion had a range of -10.0 Nm·s⁻¹ kg⁻¹ to 35.0 Nm·s⁻¹ kg⁻¹ with a statistically significant bias of 11.99 Nm·s⁻¹ kg⁻¹ (95% CI 0.004 to 0.156, p< 0.05). Bland-Altman plots for all other hip movements are in the S1 File.

## Concurrent validity of HHD versus IKD with PT measures

Bland-Altman analysis showed no significant systematic bias between measurement methods when measuring PT for hip flexion (0.08, 95%CI 0.0–0.16, $p = 0.05$), abduction (-0.09, 95%CI

**Table 2. Intersession reliability of RTD 0-100ms measurements in hip movements using a handheld dynamometer, and agreement analysis with an isokinetic dynamometer.**

| | Intersession reliability | | | | | | | | Agreement analysis | | |
|---|---|---|---|---|---|---|---|---|---|---|---|
| | Test mean(SD) session 1 (Nm·s⁻¹kg⁻¹) | Test mean(SD) session 2 (Nm·s⁻¹kg⁻¹) | Difference(SD) session 1–2 (Nm·s⁻¹kg⁻¹) | $ICC_{2,1}$ | SEM (Nm·s⁻¹kg⁻¹) | SEM % | MDC (Nm·s⁻¹kg⁻¹) | MDC % | Bias (Nm·s⁻¹kg⁻¹) | 95% LOA (Nm·s⁻¹kg⁻¹) | |
| | | | | | | | | | | Lower | Upper |
| **Hip Flexion** | 3.90 (1.53) | 3.76 (1.53) | 0.13 (1.01) | 0.85 | 0.98 | 19 | 2.18 | 42 | 8.69 | 0.0 | 17.4 |
| **Hip Extension** | 4.71 (1.87) | 4.24 (1.47) | 0.46 (1.41) | 0.77 | 0.10 | 22 | 2.24 | 50 | 0.73 | -7.0 | 8.40 |
| **Hip Abduction** | 4.06 (3.13) | 3.93 (1.56) | 0.13 (1.38) | 0.85 | 0.98 | 24 | 1.11 | 55 | 5.79 | -1.1 | 12.7 |
| **Hip Adduction** | 3.66 (1.41) | 3.99 (1.68) | -0.33 (1.16) | 0.84 | 0.82 | 21 | 1.83 | 48 | 1.98 | -1.8 | 5.08 |
| **Hip Internal rotation** | 1.80 (0.96) | 1.92 (0.80) | -0.06 (0.56) | 0.88 | 0.41 | 22 | 0.92 | 49 | 2.06 | -0.8 | 4.90 |
| **Hip External rotation** | 1.38 (0.62) | 1.51 (0.59) | -0.11 (0.47) | 0.87 | 0.36 | 19 | 0.81 | 44 | 3.90 | -0.7 | 8.50 |

SD: Standard deviation, ICC: Intraclass correlation coefficient $ICC_{2,1}$ SEM: Standard error of measurement, MDC: Minimal detectable change, Bias: mean of differences between measurement methods, 95% LOA: 95% Limits of agreement

**Table 3. Intersession reliability of RTD 0-150ms measurements in hip movements using a handheld dynamometer, and agreement analysis with an isokinetic dynamometer.**

| | Intersession reliability | | | | | | | | Agreement analysis | | |
|---|---|---|---|---|---|---|---|---|---|---|---|
| | Test mean(SD) session 1 (Nm·s$^{-1}$kg$^{-1}$) | Test mean(SD) session 2 (Nm·s$^{-1}$kg$^{-1}$) | Difference(SD) session 1–2 (Nm·s$^{-1}$kg$^{-1}$) | ICC$_{2,1}$ | SEM (Nm·s$^{-1}$kg$^{-1}$) | SEM % | MDC (Nm s$^{-1}$kg$^{-1}$) | MDC % | Bias (Nm s$^{-1}$kg$^{-1}$) | 95% LOA (Nm s$^{-1}$kg$^{-1}$) | |
| | | | | | | | | | | Lower | Upper |
| **Hip Flexion** | 4.61 (1.51) | 4.55 (1.53) | 0.07 (0.92) | 0.91 | 0.78 | 14 | 1.63 | 32 | 2.37 | -1.9 | 6.6 |
| **Hip Extension** | 5.32 (2.02) | 4.89 (1.74) | 0.43 (1.24) | 0.87 | 0.88 | 17 | 1.96 | 38 | -1.33 | -7.4 | 4.7 |
| **Hip Abduction** | 4.22 (1.87) | 4.25 (1.69) | -0.30 (1.03) | 0.91 | 0.73 | 17 | 1.63 | 39 | 3.99 | -0.5 | 12.7 |
| **Hip Adduction** | 3.75 (1.44) | 3.88 (1.60) | -0.13 (0.67) | 0.95 | 0.47 | 12 | 1.05 | 28 | 1.12 | -2.4 | 4.6 |
| **Hip Internal rotation** | 2.00 (0.89) | 2.11 (0.88) | -0.11 (0.53) | 0.90 | 0.38 | 18 | 0.84 | 41 | 1.07 | -1.3 | 3.4 |
| **Hip External rotation** | 1.92 (0.87) | 2.18 (0.87) | -0.25 (0.50) | 0.89 | 0.36 | 17 | 0.80 | 39 | 1.72 | -0.9 | 4.3 |

SD: Standard deviation, ICC: Intraclass correlation coefficient ICC$_{2,1}$ SEM: Standard error of measurement, MDC: Minimal detectable change, Bias: mean of differences between measurement methods, 95% LOA: 95% Limits of agreement

-0.23–0.05, $p$ = 0.2), internal (-0.01, 95%CI -0.13–0.1, $p$ = 0.79), and external rotation (-0.05, 95%CI -0.02–0.12, $p$ = 0.15) and therefore the two devices showed agreement. However, a significant systematic bias between measurement methods was observed when measuring PT in hip adduction (-0.22, 95%CI -0.32- -0.11, $p$ = 0.001) and extension (-0.95, 95%CI -1.2- -0.7, $p$<0.001). The limits of agreement were lowest with hip external rotation (-0.32 to 0.42 Nm·kg$^{-1}$) and highest with hip extension (-2.27–0.37 Nm·kg$^{-1}$).

A Bland-Altman plot examining agreement between the IKD and the HHD when measuring hip flexion PT is shown in Fig 4 as an example. Bland-Altman plots for all other hip movements are in the S2 File. The bias and the LOA for PT are presented in Table 5.

**Table 4. Intersession reliability of RTD 0-200ms measurements in hip movements using a handheld dynamometer, and agreement analysis with an isokinetic dynamometer.**

| | Intersession reliability | | | | | | | | Agreement analysis | | |
|---|---|---|---|---|---|---|---|---|---|---|---|
| | Test mean(SD) session 1 (Nm·s$^{-1}$kg$^{-1}$) | Test mean(SD) session 2 (Nm·s$^{-1}$kg$^{-1}$) | Difference(SD) session 1–2 (Nm·s$^{-1}$kg$^{-1}$) | ICC$_{2,1}$ | SEM (Nm·s$^{-1}$kg$^{-1}$) | SEM % | MDC (Nm·s$^{-1}$kg$^{-1}$) | MDC % | Bias (Nm·s$^{-1}$kg$^{-1}$) | 95% LOA (Nm·s$^{-1}$kg$^{-1}$) | |
| | | | | | | | | | | Lower | Upper |
| **Hip Flexion** | 4.49 (1.72) | 4.40 (1.71) | 0.08 (0.90) | 0.93 | 0.55 | 14 | 1.22 | 32 | 0.73 | 4.1 | -2.6 |
| **Hip Extension** | 5.33 (2.04) | 4.90 (1.66) | 0.42 (1.14) | 0.89 | 0.81 | 16 | 1.81 | 35 | 0.73 | -1.8 | 5.3 |
| **Hip Abduction** | 3.93 (1.57) | 3.95 (1.49) | -0.02 (0.77) | 0.93 | 0.55 | 14 | 1.22 | 31 | 1.75 | -1.8 | 5.3 |
| **Hip Adduction** | 3.35 (1.18) | 3.40 (1.39) | 0.06 (0.64) | 0.94 | 0.45 | 13 | 1.02 | 30 | 0.56 | -1.9 | 2.9 |
| **Hip Internal rotation** | 1.95 (0.80) | 2.08 (0.85) | -0.13 (0.54) | 0.88 | 0.38 | 19 | 0.85 | 42 | 0.67 | -1.1 | 2.5 |
| **Hip External rotation** | 1.86 (0.82) | 2.08 (0.83) | -0.23 (0.43) | 0.91 | 0.30 | 15 | 0.67 | 34 | 0.98 | -1.0 | 3.0 |

SD: Standard deviation, ICC: Intraclass correlation coefficient ICC$_{2,1}$ SEM: Standard error of measurement, MDC: Minimal detectable change, Bias: mean of differences between measurement methods, 95% LOA: 95% Limits of agreement

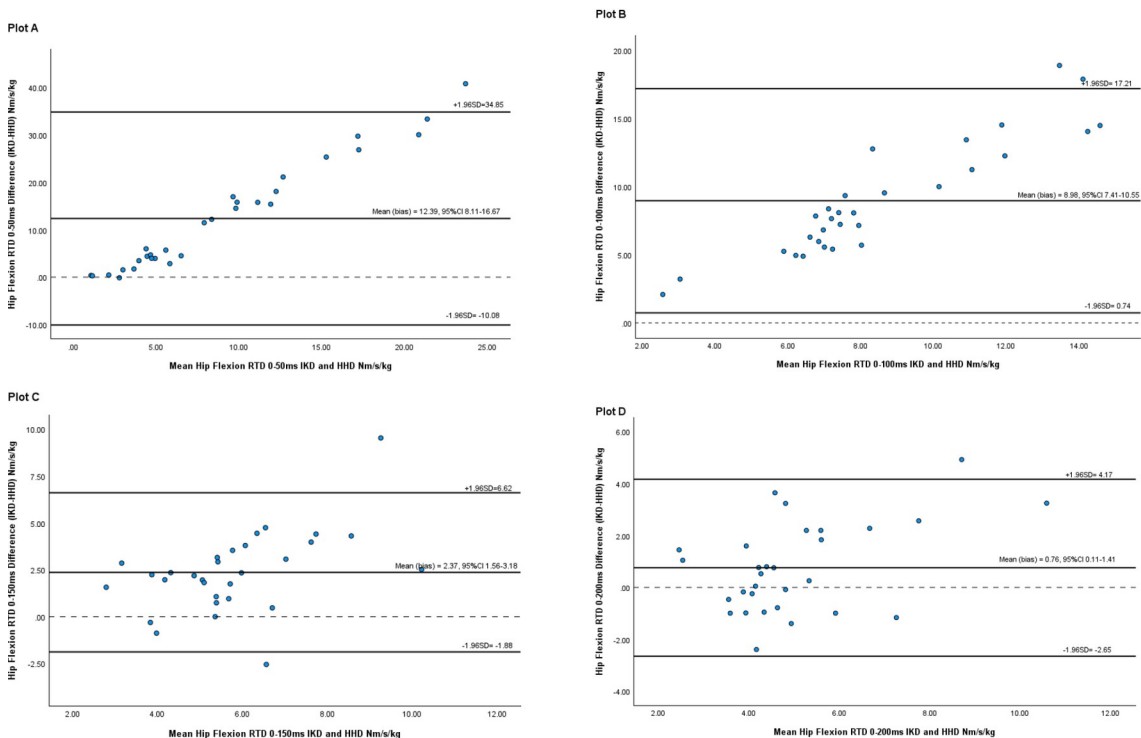

**Fig 3.** Plots A-D: Bland-Altman plots to show agreement between HHD and IKD when measuring RTD in hip flexion for group results.

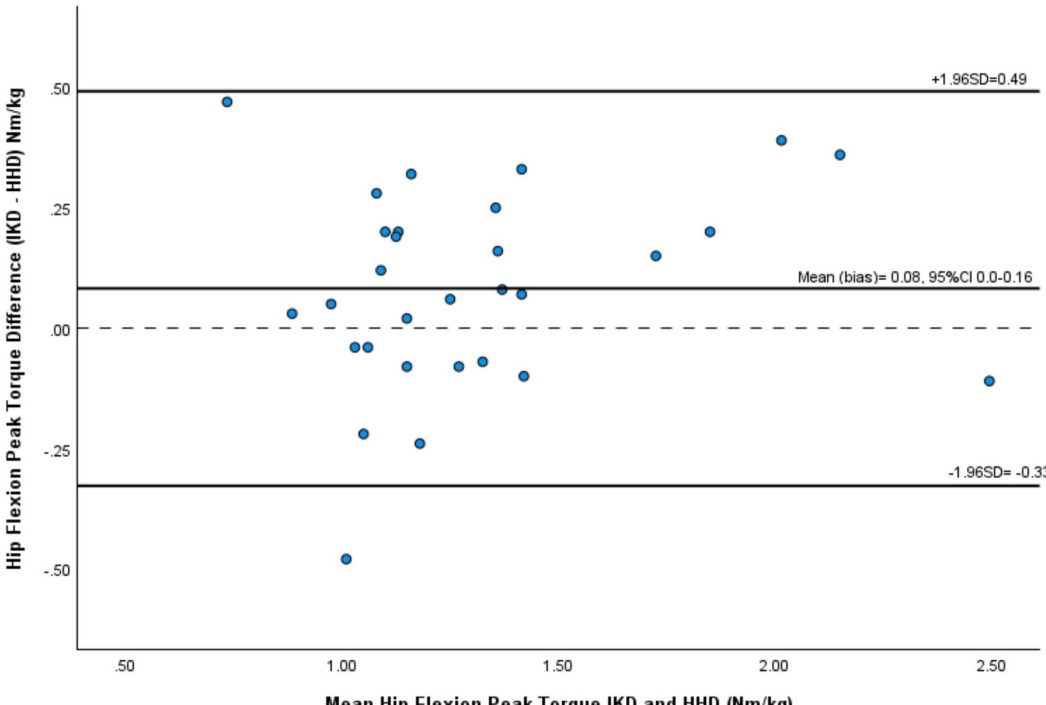

**Fig 4. Bland-Altman plot to show agreement between IKD and HHD measuring hip flexion peak torque for group results.**

## Reliability of RTD measures with HHD

Intra-rater, relative reliability results, between two test sessions, using the HHD to measure RTD in the hip movements are presented in Tables 1–4. In the six hip movements and four time epochs per movement, the relative reliability of using the HHD to measure the RTD, showed excellent or good reliability ($ICC_{2,1}$ 0.77–0.95) except in $RTD_{0-50}$ flexion ($ICC_{2,1}$ 0.72), extension ($ICC_{2,1}$ 0.5) and abduction ($ICC_{2,1}$ 0.72) which showed moderate reliability. A trend was apparent in all six hip movements showing the absolute measurement variation (SEM %) was highest in the lowest RTD time epochs, ranging from 23–36% in $RTD_{0-50}$ and 19–24% in $RTD_{0-100}$, and absolute measurement variation was lower in the higher time epochs, 12–18% in $RTD_{0-150}$, and 13–19% in $RTD_{0-200}$. The MDC% was also highest in $RTD_{0-50}$ (51–80%) and lowest in the higher time epochs $RTD_{0-150}$ (28–41%), $RTD_{0-200}$ (30–42%).

## Reliability of PT measures with HHD

The relative reliability of using the HHD to measure PT was excellent or good for all six hip movements ($ICC_{2,1}$ 0.85–0.95). Absolute measurement variation (SEM) ranged between 0.08 and 0.2 Nm·kg$^{-1}$ and as a percentage, 10–15%. MDC was lowest in hip abduction (22%) and highest in hip internal rotation (35%) (Table 5).

## Correlation of RTD with PT

A Pearson's product moment correlation revealed time epochs of $RTD_{150}$ and $RTD_{200}$ ($r$ range = .62 to .87 $p$ <0.001) correlated more strongly with PT measurements than the lowest time epoch $RTD_{50}$ ($r$ range = .23 to .76). A complete list of correlations is presented in Table 6.

## Discussion

This study demonstrates that the HHD can be used reliably to measure rate of torque development (RTD) in hip movements using a clinically feasible method. However, assessment of the concurrent validity through agreement parameters demonstrated that HHD measurements of late phase $RTD_{0-150}$ and $RTD_{0-200}$ have better agreement with the IKD than early phase $RTD_{0-50}$ and $RTD_{0-100}$. The presence of a significant systematic bias across all RTD time epochs

**Table 5. Intersession reliability of peak torque measurements in hip movements using a handheld dynamometer, and agreement analysis with an isokinetic dynamometer.**

| | Intersession reliability | | | | | | | | Agreement analysis | | |
|---|---|---|---|---|---|---|---|---|---|---|---|
| | Test mean(SD) session 1 (Nm·kg$^{-1}$) | Test mean(SD) session 2 (Nm·kg$^{-1}$) | Difference(SD) session 1–2 (Nm·kg$^{-1}$) | $ICC_{2,1}$ | SEM (Nm·kg$^{-1}$) | SEM % | MDC (Nm·kg$^{-1}$) | MDC % | Bias (Nm·kg$^{-1}$) | 95% LOA (Nm·kg$^{-1}$) | |
| | | | | | | | | | | Lower | Upper |
| **Hip Flexion** | 1.27 (0.38) | 1.29 (0.37) | -0.02 (0.26) | 0.91 | 0.18 | 14 | 0.40 | 32 | 0.08 | -0.33 | 0.50 |
| **Hip Extension** | 1.80 (0.58) | 1.72 (0.51) | 0.08 (0.30) | 0.95 | 0.20 | 12 | 0.45 | 26 | -0.95 | -2.27 | 0.37 |
| **Hip Abduction** | 1.29 (0.40) | 1.28 (0.38) | 0.15 (0.18) | 0.95 | 0.13 | 10 | 0.28 | 22 | -0.09 | -0.83 | 0.65 |
| **Hip Adduction** | 1.08 (0.29) | 1.14 (0.34) | -0.06 (0.23) | 0.85 | 0.16 | 15 | 0.36 | 32 | -0.26 | -0.76 | 0.33 |
| **Hip Internal rotation** | 0.72 (0.25) | 0.75 (0.27) | -0.03 (0.16) | 0.92 | 0.11 | 15 | 0.24 | 35 | -0.01 | -0.60 | 0.59 |
| **Hip External rotation** | 0.66 (0.21) | 0.63 (0.20) | 0.04 (0.10) | 0.94 | 0.08 | 12 | 0.17 | 26 | 0.05 | -0.32 | 0.43 |

SD: Standard deviation, ICC: Intraclass correlation coefficient $ICC_{2,1}$ SEM: Standard error of measurement, MDC: Minimal detectable change, Bias: mean of differences between measurement methods, 95% LOA: 95% Limits of agreement

**Table 6. Pearson correlation between rate of torque development and peak torque in hip movements.**

| Hip Movement | Rate of torque development time epoch | | | |
| --- | --- | --- | --- | --- |
| | 0-50ms | 0-100ms | 0-150ms | 0-200ms |
| Flexion | .76* | .83* | .62* | .81* |
| Extension | .63* | .78* | .87* | .79* |
| Abduction | .23 | .52** | .86* | .79* |
| Adduction | .43** | .70* | .74* | .81* |
| Internal Rotation | .31 | .54** | .75* | .75* |
| External Rotation | .68* | .78* | .81* | .78* |

**Correlation is significant at 0.01 level (2-tailed)

* Correlation is significant at 0.05 level (2-tailed)

indicated the HHD measurements yielded lower RTD values than the IKD and may not be considered an equivalent representation of measurements obtained with the IKD.

Peak torque (PT) measurements taken with the HHD exhibited good to excellent intra-rater, repeated reliability in hip movements and demonstrated close agreement with the IKD when measuring hip flexion, abduction, internal and external rotation.

Although the ICC had consistently high or very high values, indicating excellent reliability indices, these were accompanied by high values for SEM and MDC. This indicates that the HHD for most RTD and PT measurements have a measurement error greater than 10% and might not be able to detect clinically important changes in RTD or PT performance for hip movements.

Correlation analysis was also performed to inform clinicians that there was a high association between RTD in the later phases ($RTD_{150}$ and $RTD_{200}$) and PT.

Capturing torque measurements over extremely short time periods requires a tool sensitive enough to accurately detect subtle changes. Examining the validity of the HHD for measuring RTD in hip movements by comparing it to the IKD, and assessing their agreement, sheds light on the practical implications of this. The results from this present study suggest that the HHD RTD values in all hip movements become more consistent with those of the IKD as the contraction progresses, with late phase RTD ($RTD_{0-150}$ and $RTD_{0-200}$) demonstrating more accurate results. Whilst no other research has examined the validity across several time epochs in six hip movements, these findings are in agreement with Mentiplay et al. [10] who reported a moderate to good concurrent validity between the HHD and the IKD for $RTD_{0-200}$ in hip flexion, extension, abduction, and adduction ($r$ = .80-.92) in healthy young adults.

The larger observed differences in early phase RTD measures between the two devices are likely influenced by the discrepancy in sampling frequency, with the IKD generating five sampling points compared to the HHD's two sampling points every 50ms. While the HHD has been utilized in prior research for measuring RTD ($RTD_{0-100}$ and $RTD_{0-200}$) in individuals with femoroacetabular impingement syndrome [29] and may suffice for clinical purposes, the IKD offers the potential for greater accuracy and better representation of RTD if necessary.

The agreement analysis further highlights a statistically significant systematic difference between the two devices across all hip movements, indicating the HHD consistently yielded lower RTD values compared to the IKD. Consequently, while the results from the two devices are not interchangeable, in clinical practice where only one device is employed for assessment, achieving agreement with another device may not be as crucial.

Due to the limited literature specifically examining RTD measurements between dynamometers, the clinical consequences of bias in a particular direction were not predefined.

Instead, trends in agreement were identified, noting that later-phase RTD measurements exhibited lower bias and narrower confidence intervals. While establishing clinically meaningful thresholds for bias is important, valuable insights into the agreement between measurements from HHDs and IKDs are still provided by the findings from this present study. Now that it is understood the HHD consistently measured lower than IKD, these findings can help inform clinical decision-making and guide future research on setting acceptable levels of bias. Further research is needed to help define acceptable limits of agreement in RTD measurements between dynamometers.

Examination of individual hip movements' agreement results at $RTD_{0-150}$ and $RTD_{0-200}$ reveal that hip abduction exhibited the most significant systematic bias between the dynamometers. This discrepancy is likely attributable to differences in participant positioning. The selected position aimed to enhance set up rigidity, crucial for RTD assessment, by optimizing participant stabilization and belt fixation to the physiotherapy plinth. Previous research demonstrated reduced error levels (SEM%) in the supine position (2.9%) compared to side lying (8.5%) when measuring PT [30]. While both Mentiplay et al. [10] and Ishøi et al. [11] also utilized the supine position in their RFD studies using the HHD, replicating this setup on the IKD in this current study was unfeasible due to its pre-programmed side lying configuration. Although the hip joint angle remains consistent with the two positions, further research has indicated that side lying tends to yield higher PT values than supine positioning [31, 32]. Consequently, this disparity likely contributed to the observed increase in systematic difference seen in the agreement results for hip abduction (bias $RTD_{0-200}$ 1.75 $Nm \cdot s^{-1} kg^{-1}$). However, repeated reliability measures for the supine position using the HHD are excellent (ICC 0.93) providing clinicians with confidence in this position for repeated measures whilst being mindful that it may produce lower absolute values than side lying.

Conversely for PT, there was close agreement between the two devices for hip flexion, abduction, internal and external rotation as the bias was low and 95% CIs included 0. As with RTD, dissimilar testing positions potentially attributed to significant systematic bias observed for hip extension and adduction with the HHD yielding higher PT measurements compared to the IKD. Despite other research exploring various positions when measuring PT with a HHD in hip movements, a universally agreed-upon best position remains lacking [33]. The position used for hip extension is variable in other studies. Bazett-Jones et al. [21] reported poor concurrent validity between the HHD and IKD ($r = 0.19$) for hip extension PT when the identical position was used for both methods, aligning with the supine position at a 0˚ hip joint angle used in this study for the IKD. In contrast, Mentiplay et al. [10] adopted the prone hip extension position matching the position used for the HHD in this study and observed strong concurrent validity between the two methods. Hence, it may be advisable for future studies to consider the prone hip extension position for improved validity.

Good to excellent intra-rater repeated reliability was demonstrated for RTD and PT with the HHD across all hip movements, consistent with findings from other studies [10, 11, 15]. More specifically, reliability levels of RTD improved with larger time epochs, exhibiting lower absolute measurement variation (SEM) thereby indicating that $RTD_{0-150}$ and $RTD_{0-200}$ are better suited for clinical measurement. These findings align with previous research involving healthy populations [10, 34].

Comparison of RTD and PT measurement variation using the HHD reveals that RTD measures exhibit greater variability than PT, consistent with previous research [10, 11]. Consequently, RTD measures are less precise, making it challenging to detect small but potentially clinically significant changes.

Variation in SEM levels may also be influenced by participant characteristics and differing populations. In this present study, SEM values for PT ranged from 10–15% which are similar

[15, 35] or higher than those reported in previous studies using the HHD [10, 11]. SEM levels $RTD_{0-200}$ for hip flexion (14%), extension (16%), abduction (14%) and adduction (13%) are similar or slightly higher compared to those reported by Ishøi et al. [11], who found SEM levels for $RTD_{0-200}$ in hip flexion (7.4%), extension (12.3%), abduction (13.8%) and adduction (14%) (age 25.4 ± 4.2 years). Similarly, Mentiplay et al. [10] using two assessors, reported lower SEM values for $RTD_{0-200}$, with hip flexion ranging from 9.65% to 11.68%, extension from 9.70% to 10.71%, abduction from 13.08% to 15.69% and adduction from 11.06–13.0% (age 22.87±5.08 years). The higher variation in this study could be attributed to a broader age range and higher mean age (30 ± 8 years), along with greater heterogeneity in height and weight among subjects, resulting in increased variability.

These results suggest that the HHD may not reliably detect changes at the previously suggested 10% threshold considered clinically relevant for assessing muscle force improvements or deteriorations [36, 37]. However, in clinical populations such as individuals with hip osteoarthritis, the SEM for assessing peak force with the HHD was found to be lower (2.12–8.49%) [38] indicating the HHD may offer a more reliable assessment method in clinical populations, sensitive enough for detecting clinically significant changes.

Clinically useful MDC% values for PT measurements in hip flexion, extension, abduction, and adduction using the HHD in the current study (22–35) are higher compared to those reported by Ishøi et al. [11] (MDC% 16–21%) and Mentiplay et al. [10] (MDC% 16–27%). However, individuals with hip pain can exhibit substantial deficits in peak force/torque ranging from 16–28% [17, 39–42] compared to healthy individuals, suggesting that peak force/torque changes in patients undergoing rehabilitation could potentially surpass these MDC values.

RTD MDC results are notably higher ($RTD_{0-200}$ 30–35%) than PT values which has previously been observed 20–39% [11] and 27–43% [10]. Research is limited but significant RTD deficits have been observed within clinical populations with reduced RTD observed in hip abductors and extensors of females with patellofemoral pain (33–51%) [43] and 61–71% greater RTD in hip abduction and adduction in younger healthy women (age 21.8 ±2.1years) compared to older women fallers (> 60years) [44].

This study revealed strong positive correlation between late phase RTD and PT in hip joint movements which has not previously been examined. Similarly late phase RTD correlated with PT in quadriceps and hamstring muscles [6]. In the context of rehabilitation, this correlation suggests that interventions aimed at increasing PT may concurrently improve late-phase RTD. Such insights are valuable for clinicians, as targeted rehabilitation commonly includes exercises to enhance PT and mitigate deficits.

## Study limitations

The study had limitations, including the non-randomized order of hip movement testing to prevent individual muscles from being tested succinctly with differing hip movements. Additionally, conducting test sessions on the same day, rather than on different days as is typical in clinical settings, may have introduced fatigue potentially influencing the outcomes, and affecting the generalizability of the findings. However, the protocol was defined to reduce time demands on participants during the Covid-19 pandemic. While the consistent test order minimized variability between groups, we mitigated fatigue concerns by ensuring rest time between contractions and incorporating additional rest periods during transitions between different movements. Participant feedback and observations of fatigue levels guided the need for longer rest periods when warranted.

Another limitation was differing positions between the HHD and the IKD. While joint angles remained consistent for both dynamometers, the HHD participant positions were

chosen for clinical feasibility as opposed to the pre-programmed and standardized positions of the IKD, but this likely had a negative impact on the agreement results.

Lastly, the method chosen for calculating the onset of torque production for RTD aimed to achieve the highest possible precision [4]. However, this method is time-consuming and may not be practical in a clinical setting. To address this, an automatic force onset threshold can be set directly on the HHD, simplifying the process. Notably, the post-test calculations required to determine RTD values may present a barrier to RTD measurement in clinical practice. To address this, the development of accurate, automatic conversion within the HHD software is recommended. Alternatively, external software programmes can be used [10].

## Conclusion

This study offers a comprehensive analysis in healthy subjects for RTD and PT using a HHD in hip movements and provides comparative analysis with measurements taken using an IKD. Results indicate that late phase $RTD_{0-150}$, and $RTD_{0-200}$, in all hip movements exhibit better reliability, lower SEM%, lower MDC%, and improved agreement with the IKD compared to early phase $RTD_{0-50}$. Therefore, these late phase measurements are recommended for use with the HHD.

RTD measurement with a belt stabilized HHD in hip movements (four trials of MVIC with the instruction to push as 'fast and hard' as possible to encourage rapid muscle contraction) is a novel approach, yet it offers a method that is accessible for clinical application. However, clinicians should be mindful of the differences in setup between the HHD and IKD, as these variations can impact validity and reliability.

## Supporting information

**S1 Checklist. Human participants research checklist.**
(DOCX)

**S1 File. S1 Appendix: Bland-Altman plots to show agreement of HHD with IKD measuring RTD in hip movements for group results.**
(DOCX)

**S2 File. S2 Appendix: Bland-Altman plots to show agreement of HHD with IKD measuring peak torque in hip movements for group results.**
(DOCX)

## Author Contributions

**Conceptualization:** Katherine McNabb, James Selfe, Neil D. Reeves, Michael Callaghan.

**Data curation:** Katherine McNabb.

**Formal analysis:** Katherine McNabb.

**Investigation:** Katherine McNabb.

**Methodology:** Katherine McNabb, Neil D. Reeves, Michael Callaghan.

**Project administration:** Katherine McNabb.

**Resources:** Katherine McNabb.

**Software:** Katherine McNabb.

**Supervision:** María B. Sánchez, James Selfe, Neil D. Reeves, Michael Callaghan.

**Writing – original draft:** Katherine McNabb, James Selfe, Neil D. Reeves, Michael Callaghan.

**Writing – review & editing:** Katherine McNabb, María B. Sánchez, Neil D. Reeves, Michael Callaghan.

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
