## [Decision Letter · Decision Letter 0]

24 Mar 2024

PONE-D-24-01588Handheld dynamometry: validity and reliability of measuring hip joint rate of torque development and peak torque.PLOS ONE

Dear Dr. McNabb,

Thank you for submitting your manuscript to PLOS ONE. After careful consideration, we feel that it has merit but does not fully meet PLOS ONE’s publication criteria as it currently stands. Therefore, we invite you to submit a revised version of the manuscript that addresses the points raised during the review process.

The reviewers have provided some useful comments that can assist the manuscript to improve. They both find the study of potential use, an evaluation I agree with. You will see both have raised some concerns and have made suggestions, I would encourage you to carefully consider and address . Areas that have been raised by both reviewers, such a s residual fatigue or 'analytical goals', as well as any other methodological aspects, should be of particular attention.

We look forward to receiving your revised manuscript.

Kind regards,

Theodoros M. Bampouras

Academic Editor

PLOS ONE

Journal Requirements:

2. Please amend the manuscript submission data (via Edit Submission) to include author James Selfe.

Reviewers' comments:

Reviewer's Responses to Questions

**Comments to the Author**

1. Is the manuscript technically sound, and do the data support the conclusions?

Reviewer #1: Partly

Reviewer #2: Partly

2. Has the statistical analysis been performed appropriately and rigorously? 

Reviewer #1: Yes

Reviewer #2: Yes

3. Have the authors made all data underlying the findings in their manuscript fully available?

Reviewer #1: No

Reviewer #2: Yes

4. Is the manuscript presented in an intelligible fashion and written in standard English?

Reviewer #1: Yes

Reviewer #2: Yes

5. Review Comments to the Author

Reviewer #1: In this study, the authors aimed at investigating the concurrent validity of handheld (HHD) and isokinetic (IKD) dynamometers, as well as the intrarater reliability of the HHD when measuring hip muscle peak torque (PT) and rate of torque development (RTD) outputs. This is a very relevant topic and I commend the authors for trying their endeavor. Even though, major concerns limit the interest in recommending this study for publication.

My main concern is related to the number of tests performed. According to the procedures, each muscle group has been tested at least 24 times (Session 1: 4 trials to measure PT using IKD, 4 trials to measure RTD using IKD; Session 2: 4 trials to measure PT using HHD, 4 trials to measure RTD using HHD plus re-tests). Some of them may have been tested even more, as some of the hip muscles can be activated in more than one movement. In addition, the sessions were about 30-minutes apart, with the second one including test and retest measures. Although participants of this study were free from pain or injury, it is important to consider that, in this scenario, fatigue may have played a role in the results’ interpretation. Another important aspect that should be considered is the physical activity level of participants (different effects of fatigue are expected between physically active and sedentary ones). However, no information about physical activity levels or perceived efforts during testing has been provided.

Why analyses comparing PT and IKD have not been performed as previously (10.1371/journal.pone.0140822)? They are important and can further support the agreement analyses.

The discussion of this study is too long and hard to follow. Short paragraphs like “The present study demonstrated that late phase RTD0-150 and RTD0-200 measurements are more reliable, have better agreement with the IKD, and relate to PT, than early phase RTD0-50 and RTD0-100.” are common and does not bring a deeper discussion considering previous literature, current findings, and overall implications.

In the limitations, the authors have stated that “Lastly, the study employed a labour-intensive onset of contraction calculation method for RTD which may not be practical for clinical practice due to its time-consuming nature.” and this contrasts with the purpose of the study. What clinicians should do about RTD measures after reading this manuscript? Mentiplay et al. (10.1371/journal.pone.0140822) have provided software program that helps clinicians to extract RTD values when using a similar protocol to them. Why something similar has not been considered?

The introduction of this study should be better aligned with its aims. An example of that is the focus on the importance of testing maximal strength in patients with hip related pain, whereas only pain or injury-free participants have been included.

The authors should be clearer about the need for correlation analyses between PT and RTD measures. I suggest the inclusion of information in the introduction. Also, in the discussion, the authors have stated that “Correlation analysis was also performed to inform clinicians that there was a high association between RTD in the later phases (RTD150 and RTD200) and PT. This indicates that the simpler PT calculation can be used in clinical practice.” and I wonder if the results of this study support such a statement. Does this suggest that clinicians should use PT measures to surrogate RTD ones? How would that work and how they should be interpreted?

The higher SEM and MDC reported by the authors may be explained by the inclusion of participants who were free from pain or injury (10.1371/journal.pone.0140822). Can the authors use previous literature including other samples (e.g., samples with hip related pain) to provide further information of how clinicians can interpret absolute reliability?

I understand how hard is to perform IKD and PT protocols using the same testing position. However, why positions had to be different for hip abd/add? Lower values of hip abd, for instance, may be expected when testing in supine as compared to side-lying (10.1016/j.ptsp.2020.01.005; 10.1016/j.jbmt.2022.02.011; 10.2519/jospt.2023.12045). This may have influenced differences/disagreements between instruments.

Thank you for the opportunity to review this manuscript!

Reviewer #2: Proposed additional limitations:

- Before conducting the study, it was essential to make an informed decision about the clinical importance of any potential bias between the two instruments (IKD and HHD). Without such decision, the presence of a statistically significant bias between the two instruments would present a dilemma similar to interpreting p-values and effect size. For example, while a bias of >10 (Nm/s/kg or Nm/kg) could be considered large enough to inform meaningful bias, this may not be the case for any bias <1 (Nm/s/kg or Nm/kg). This may therefore, affect interpretation of the findings and importance of the found significantly different biases.

- Data collection was completed in one day due to restrictions during the Covid era, with approximately 30-minute delay between measurements 1 and 2. This will not directly translate to delays practiced between repeat measurements in a clinical setting, potentially affecting generalisation of found measurement repeatability.

- Repeatability of measurements may be lower in clinical populations due to changing health states and pain levels. It would be insightful to explore/speculate whether a more conservative multiple of the MDC is needed to accurately capture these changes when using HHD to assess treatment responses.

Methodological points:

- Consideration should be given to the accurate reporting of sampling frequency of IKD, which is typically 100 Hz but was sampled by the Labchart data acquisition system at a different (1000 Hz) frequency. Did IKD have actual SF of 1000 Hz?

- Clarification is needed on the duration of time under tension during MVIC for completion of the method, and how peak torque was determined, particularly regarding data manipulation: according to the manuscript the highest peak was reported over 4 trials: i.e. this was the maximum (one data point) value obtained out of 4 trials.

- Confirmation is required on whether the moment arm was measured once or twice during HHD.

- The method for detecting onset of contraction needs clarification, particularly regarding the translation of (3) data points into different time windows due to differences in sampling frequency between instruments.

- You have used “onset of contraction” for referring to the onset of torque production (Data Extraction). To prevent confusion, it might be better to reserve “contraction” to refer to muscle activity. Muscle activity starts earlier than increase in torque, and no EMG was used in this study

Results:

I would have liked to see the results of t-test when reporting significant Bias in relevant tables. I am aware of space limitation.

6. PLOS authors have the option to publish the peer review history of their article (what does this mean?). If published, this will include your full peer review and any attached files.

Reviewer #1: No

Reviewer #2: No

---

## [Author Response · Author response to Decision Letter 0]

5 Jun 2024

Comments from the editors and reviewers:

We would like to thank the Editor and the Reviewers for their comments, observations and suggestions. Please find our responses below where we hope to have addressed all these elements. Where these generated changes in the manuscript, the correct location is indicated and is highlighted within the manuscript’s text. An unmarked version of the revised paper without tracked changes has also been submitted.

Reviewer comment Response

1.Reviewer #1

My main concern is related to the number of tests performed. According to the procedures, each muscle group has been tested at least 24 times (Session 1: 4 trials to measure PT using IKD, 4 trials to measure RTD using IKD; Session 2: 4 trials to measure PT using HHD, 4 trials to measure RTD using HHD plus re-tests). Some of them may have been tested even more, as some of the hip muscles can be activated in more than one movement. In addition, the sessions were about 30-minutes apart, with the second one including test and retest measures. Although participants of this study were free from pain or injury, it is important to consider that, in this scenario, fatigue may have played a role in the results’ interpretation.

 Thank you for your comment regarding the number of tests performed and the potential impact of fatigue on the interpretation of our results. The decision to have both test sessions on the same day rather than separate days was forced on us when Covid-19 restrictions were imposed at our institution. We too were concerned about the effect of fatigue on the performance and results from this more intensive testing protocol.

To mitigate this concern, we implemented measures to closely monitor peak torque values across the four tests for each movement. Thus, visual monitoring was conducted to detect any trends indicating fatigue evidenced by a sequential reduction in peak torque measures. Rest time between contractions was strictly enforced, and additional rest periods were incorporated during transitions between different movements. Moreover, participant feedback and observations of fatigue levels throughout the sessions were carefully considered to identify instances where longer rest periods were warranted.

The results were also used to assess the presence of fatigue. For analysis, we selected the highest measure out of the four recorded for each test. However, it is noteworthy that this measure varied in its position across the four tests. If fatigue had been evident during the series of four tests, we would have anticipated a consistent occurrence of lower values occurring later in the sequence of tests.

Additionally, we acknowledge that muscle activation overlap in certain movements, could potentially resulting in repeated testing of specific muscle groups. To address this concern, the order of positions was chosen to ensure that tests engaging the same muscle groups were not conducted sequentially. 

We revisited the data for internal rotation movement using the HHD. For example, internal rotation could potentially demonstrate fatigue due to two factors. Firstly, the movement engages muscles that may have already been fatigued by prior movements, such as abduction, involving the gluteus medius, TFL, and gluteus minimus. Secondly, internal rotation was positioned as the penultimate movement in our protocol, implying that fatigue resulting from the duration of testing could reasonably be expected. 

To examine this in the data, we computed the average of the highest three peak torque measurements for hip internal rotation during both test sessions utilizing the HHD. We then calculated the percentage difference between these sessions. Interestingly, only 7 out of the 30 participants exhibited a reduction in peak torque during the second session compared to the first, where 6 of these demonstrated a decline in peak torque exceeding 10%. These findings suggest that a discernible trend of fatigue across the tests is not evident. 

Considering your comment, we have included it as a study limitation in the manuscript addressing the potential influence of fatigue on our results and the measures taken to mitigate its effects. We have also added text to acknowledge repeated muscle group testing. 

The manuscript ‘Study limitations’ section, now reads:

‘The study had limitations, including the non-randomized order of hip movement testing to prevent individual muscles from being tested succinctly with differing hip movements. Additionally, conducting test sessions on the same day, rather than on different days as is typical in clinical settings, may have introduced fatigue potentially influencing the outcomes, and affecting the generalizability of the findings. However, the protocol was defined to reduce time demands on participants during the Covid-19 pandemic. While the consistent test order minimized variability between groups, we mitigated fatigue concerns by ensuring rest time between contractions and incorporating additional rest periods during transitions between different movements. Participant feedback and observations of fatigue levels guided the need for longer rest periods when warranted. 

2.Reviewer #1

Another important aspect that should be considered is the physical activity level of participants (different effects of fatigue are expected between physically active and sedentary ones). However, no information about physical activity levels or perceived efforts during testing has been provided.

 We appreciate your suggestion regarding the consideration of participants' physical activity levels, as it can indeed influence the interpretation of our findings, especially in relation to fatigue effects. In fact, we did collect baseline activity data using the International Physical Activity Questionnaire (IPAQ)-short form. Whilst we intended using it to compare this healthy population with patient populations, we agree that including IPAQ data could provide valuable insights into the fitness capacity of the healthy population and so the data has now been included in the results section. 

In the manuscript ‘Methods’ section, under ‘Experimental procedure and design’ subheading, the relevant part now reads: 

'Immediately prior to the first test session anthropometric measurements were taken and the International Physical Activity Questionnaire (IPAQ), short form was completed to establish participants physical activity levels.’

And in the manuscript ‘Result’s section it reads: 

‘All 30 participants data were used in the statistical analysis (mean age = 30 ± 8 years; height = 1.70 ± 0.08m; mass =70.55 ± 11.76kg), with 13 males included in the study. IPAQ results classified individuals as participating in low (6), moderate (11) and high (13) activity levels.’ 

3.Reviewer #1

Why analyses comparing PT and IKD have not been performed as previously (10.1371/journal.pone.0140822)? They are important and can further support the agreement analyses.

We were a little uncertain of the exact meaning of the reviewer’s comment here but have done our best to answer in line with our understanding of the reviewer’s question. We have examined the suggested research and have two potential interpretations. If the reviewer is asking why has agreement between HHD and the IKD when measuring the peak torque not been performed? This analysis had already been performed and was included in the results section and table 5. 

On the other hand, if the reviewer is asking why ICC analysis has not been performed to assess concurrent validity of the HHD and IKD for PT and RTD, then this is because we wanted to give greater focus to the agreement analysis. ICC does not distinguish between bias (systematic differences) and variability (random error) between measurements. Therefore, it can be difficult to determine the source of the disagreement between devices and our preferred analysis was Bland and Altman. 

4.Reviewer #1 

The discussion of this study is too long and hard to follow. 

Short paragraphs like “The present study demonstrated that late phase RTD0-150 and RTD0-200 measurements are more reliable, have better agreement with the IKD, and relate to PT, than early phase RTD0-50 and RTD0-100.” are common and does not bring a deeper discussion considering previous literature, current findings, and overall implications.

We thank the reviewer for this comment as it enabled us to reconsider and re-write much of the discussion section. Given the extensive and interesting findings, we have deepened our analysis in some of the most important points, particularly emphasizing the clinical relevance of our results. We have highlighted the intrinsic differences between PT and RTD and their measurement, aiming to make the information both useful and interesting to a clinical audience. 

The manuscript ‘Discussion’ section reads as follows.

‘This study demonstrates that the HHD can be used reliably to measure rate of torque development (RTD) in hip movements using a clinically feasible method. However, assessment of the concurrent validity through agreement parameters demonstrated that HHD measurements of late phase RTD0-150 and RTD0-200 have better agreement with the IKD than early phase RTD0-50 and RTD0-100. The presence of a significant systematic bias across all RTD time epochs indicated the HHD measurements yielded lower RTD values than the IKD and may not be considered an equivalent representation of measurements obtained with the IKD. 

Peak torque (PT) measurements taken with the HHD exhibited good to excellent intra-rater, repeated reliability in hip movements and demonstrated close agreement with the IKD when measuring hip flexion, abduction, internal and external rotation.

Although the ICC had consistently high or very high values, indicating excellent reliability indices, these were accompanied by high values for SEM and MDC. This indicates that the HHD for most RTD and PT measurements have a measurement error greater than 10% and might not be able to detect clinically important changes in RTD or PT performance for hip movements.

Correlation analysis was also performed to inform clinicians that there was a high association between RTD in the later phases (RTD150 and RTD200) and PT.

Capturing torque measurements over extremely short time periods requires a tool sensitive enough to accurately detect subtle changes. Examining the validity of the HHD for measuring RTD in hip movements by comparing it to the IKD, and assessing their agreement, sheds light on the practical implications of this. The results from this present study suggest that the HHD RTD values in all hip movements become more consistent with those of the IKD as the contraction progresses, with late phase RTD (RTD0-150 and RTD0-200) demonstrating more accurate results. Whilst no other research has examined the validity across several time epochs in six hip movements, these findings are in agreement with Mentiplay et al (2015) who reported a moderate to good concurrent validity between the HHD and the IKD for RTD0-200 in hip flexion, extension, abduction, and adduction (r= .80-.92) in healthy young adults.

The larger observed differences in early phase RTD measures between the two devices are likely influenced by the discrepancy in sampling frequency, with the IKD generating five sampling points compared to the HHD’s two sampling points every 50ms. While the HHD has been utilized in prior research for measuring RTD (RTD0-100 and RTD0-200) in individuals with femoroacetabular impingement syndrome (Ishøi et al., 2021) and may suffice for clinical purposes, the IKD offers the potential for greater accuracy and better representation of RTD if necessary.

The agreement analysis further highlights a statistically significant systematic difference between the two devices across all hip movements, indicating the HHD consistently yielded lower RTD values compared to the IKD. Consequently, while the results from the two devices are not interchangeable, in clinical practice where only one device is employed for assessment, achieving agreement with another device may not be as crucial. 

Examination of individual hip movements’ agreement results at RTD0-150 and RTD0-200 reveal that hip abduction exhibited the most significant systematic bias between the dynamometers. This discrepancy is likely attributable to differences in participant positioning. The selected position aimed to enhance set up rigidity, crucial for RTD assessment, by optimizing participant stabilization and belt fixation to the physiotherapy plinth. Previous research demonstrated reduced error levels (SEM%) in the supine position (2.9%) compared to side lying (8.5%) when measuring PT (Thorborg et al., 2010a). While both Mentiplay et al (2015) and Ishoi et al (2019b) also utilized the supine position in their RFD studies using the HHD, replicating this setup on the IKD in this current study was unfeasible due to its pre-programmed side lying configuration. Although the hip joint angle remains consistent with the two positions, further research has indicated that side lying tends to yield higher PT values than supine positioning (Widler et al., 2009; Waiteman et al., 2023). Consequently, this disparity likely contributed to the observed increase in systematic difference seen in the agreement results for hip abduction (bias RTD0-200 1.75 Nm·s-1 kg-1). However, repeated reliability measures for the supine position using the HHD are excellent (ICC 0.93) providing clinicians with confidence in this position for repeated measures whilst being mindful that it may produce lower absolute values than side lying. 

Conversely for PT, there was close agreement between the two devices for hip flexion, abduction, internal and external rotation as the bias was low and CIs included 0. As with RTD, dissimilar testing positions potentially attributed to significant systematic bias observed for hip extension and adduction with the HHD yielding higher PT measurements compared to the IKD. Despite other research exploring various positions when measuring PT with a HHD in hip movements, a universally agreed-upon best position remains lacking (Stark et al., 2011). The position used for hip extension is variable in other studies. Bazett-Jones et al (2020) reported poor concurrent validity between the HHD and IKD (r = 0.19) for hip extension PT when the identical position was used for both methods, aligning with the supine position at a 0° hip joint angle used in this study for the IKD. In contrast, Mentiplay et al (2015) adopted the prone hip extension position matching the position used for the HHD in this study and observed strong concurrent validity between the two methods. Hence, it may be advisable for future studies to consider the prone hip extension position for improved validity.

Good to excellent intra-rater repeated reliability was demonstrated for RTD and PT with the HHD across all hip movements, consistent with findings from other studies (Mentiplay et al., 2015; Martins et al., 2017; Ishøi, Hölmich and Thorborg, 2019b). More specifically, reliability levels of RTD improved with larger time epochs, exhibiting lower absolute measurement variation (SEM) thereby indicating that RTD0-150 and RTD0-200 are better suited for clinical measurement. These findings align with previous research involving healthy populations (Mentiplay et al., 2015; Gonçalves et al., 2022).

Comparison of RTD and PT measurement variation using the HHD reveals that RTD measures exhibit greater variability than PT, consistent with previous research (Mentiplay et al., 2015; Ishøi, Hölmich and Thorborg, 2019b). Consequently, RTD measures are less precise, making it challenging to detect small but potentially clinically significant changes. 

Variation in SEM levels may also be influenced by participant characteristics and differing populations. In this present study, SEM values for PT ranged from 10-15% in the current study which are similar (Martins et al., 2017; Florencio et al., 2019) or higher than those reported in previous studies using the HHD (Mentiplay et al., 2015; Ishøi, Hölmich and Thorborg, 2019b). SEM levels RTD0-200 for hip flexion (14%), extension (16%), abduction (14%) and adduction (13%) are similar or slightly higher compared to those reported by Ishøi et al.

---

## [Editor Report · Decision Letter 1]

4 Jul 2024

PONE-D-24-01588R1Handheld dynamometry: validity and reliability of measuring hip joint rate of torque development and peak torque.PLOS ONE

Dear Dr. McNabb,

Thank you for submitting your manuscript to PLOS ONE. After careful consideration, we feel that it has merit but does not fully meet PLOS ONE’s publication criteria as it currently stands. Therefore, we invite you to submit a revised version of the manuscript that addresses the points raised during the review process.

 The authors are commended for the thorough and extensive revision of the manuscript. Please include a version (it can even be a shortened one) of your reply to R2 re the 'the clinical importance of any potential bias' in the manuscript, so the reader is explicitly informed about the difficulty in determining that. Additionally, please standardise reporting as either 'CI' or '95%CI' (preferable).

We look forward to receiving your revised manuscript.

Kind regards,

Theodoros M. Bampouras

Academic Editor

PLOS ONE
---

## [Author Response · Author response to Decision Letter 1]

19 Jul 2024

Comments from the editors and reviewers:

We would like to thank the Editor and the Reviewers for their comments, observations and suggestions. Please find our responses below where we hope to have addressed all these elements. Where these generated changes in the manuscript, the correct location is indicated and is highlighted within the manuscript’s text. An unmarked version of the revised paper without tracked changes has also been submitted.

1.Please include a version (it can even be a shortened one) of your reply to R2 re the 'the clinical importance of any potential bias' in the manuscript, so the reader is explicitly informed about the difficulty in determining that. 

‘Thank you for your comment. We have inserted a paragraph into the discussion section to address this. This section in the manuscript ‘Discussion’ reads as follows.

The agreement analysis further highlights a statistically significant systematic difference between the two devices across all hip movements, indicating the HHD consistently yielded lower RTD values compared to the IKD. Consequently, while the results from the two devices are not interchangeable, in clinical practice where only one device is employed for assessment, achieving agreement with another device may not be as crucial. 

Due to the limited literature specifically examining RTD measurements between dynamometers, the clinical consequences of bias in a particular direction were not predefined. Instead, trends in agreement were identified, noting that later-phase RTD measurements exhibited lower bias and narrower confidence intervals. While establishing clinically meaningful thresholds for bias is important, valuable insights into the agreement between measurements from HHDs and IKDs are still provided by the findings from this present study. Now that it is understood the HHD consistently measured lower than IKD, these findings can help inform clinical decision-making and guide future research on setting acceptable levels of bias. Further research is needed to help define acceptable limits of agreement in RTD measurements between dynamometers.’

2.Additionally, please standardise reporting as either 'CI' or '95%CI' (preferable).

This has been updated to 95%CI throughout the manuscript, including on the Bland and Altman plots. Changes have been highlighted.

---

## [Decision Letter · Decision Letter 2]

2 Aug 2024

Handheld dynamometry: validity and reliability of measuring hip joint rate of torque development and peak torque.

PONE-D-24-01588R2

Dear Dr. McNabb,

We’re pleased to inform you that your manuscript has been judged scientifically suitable for publication and will be formally accepted for publication once it meets all outstanding technical requirements.

Kind regards,

Benjamin F Mentiplay, PhD

Academic Editor

PLOS ONE

Additional Editor Comments (optional):

Reviewers' comments:

Reviewer's Responses to Questions

**Comments to the Author**

1. If the authors have adequately addressed your comments raised in a previous round of review and you feel that this manuscript is now acceptable for publication, you may indicate that here to bypass the “Comments to the Author” section, enter your conflict of interest statement in the “Confidential to Editor” section, and submit your "Accept" recommendation.

Reviewer #1: All comments have been addressed

Reviewer #2: All comments have been addressed

2. Is the manuscript technically sound, and do the data support the conclusions?

Reviewer #1: Yes

Reviewer #2: Yes

3. Has the statistical analysis been performed appropriately and rigorously? 

Reviewer #1: Yes

Reviewer #2: Yes

4. Have the authors made all data underlying the findings in their manuscript fully available?

Reviewer #1: Yes

Reviewer #2: Yes

5. Is the manuscript presented in an intelligible fashion and written in standard English?

Reviewer #1: Yes

Reviewer #2: Yes

6. Review Comments to the Author

Reviewer #1: Thank you for the opportunity to review the revised version of this manuscript. The authors have addressed all the comments and I commend them for their endeavor.

Reviewer #2: I have now reviewed responses of the authors to the paints I raised and feel that paper is greatly improved and can be used with cautious reader who note limitations of the study.

7. PLOS authors have the option to publish the peer review history of their article (what does this mean?). If published, this will include your full peer review and any attached files.

Reviewer #1: No

Reviewer #2: No

---

## [Editor Report · Acceptance letter]

7 Aug 2024

PONE-D-24-01588R2 

PLOS ONE

Dear Dr. McNabb, 

I'm pleased to inform you that your manuscript has been deemed suitable for publication in PLOS ONE. Congratulations! Your manuscript is now being handed over to our production team.

Kind regards, 

on behalf of

Dr. Benjamin F Mentiplay 

Academic Editor

PLOS ONE